# Association between Urinary Metabolites and the Exposure of Intensive Care Newborns to Plasticizers of Medical Devices Used for Their Care Management

**DOI:** 10.3390/metabo11040252

**Published:** 2021-04-19

**Authors:** Lise Bernard, Yassine Bouattour, Morgane Masse, Benoît Boeuf, Bertrand Decaudin, Stéphanie Genay, Céline Lambert, Emmanuel Moreau, Bruno Pereira, Jérémy Pinguet, Damien Richard, Valérie Sautou

**Affiliations:** 1Université Clermont Auvergne, Clermont Auvergne INP, CNRS, CHU Clermont Ferrand, ICCF, F-63000 Clermont-Ferrand, France; lise.bernard@uca.fr (L.B.); ybouattour@chu-clermontferrand.fr (Y.B.); 2Université de Lille, CHU Lille, ULR 7365 GRITA, F-59000 Lille, France; morgane.masse@univ-lille.fr (M.M.); bertrand.decaudin@univ-lille.fr (B.D.); stephanie.genay@univ-lille.fr (S.G.); 3CHU Clermont-Ferrand, Service Réanimation Pédiatrique et Périnatalogie, F-63000 Clermont-Ferrand, France; bboeuf@chu-clermontferrand.fr; 4CHU Clermont-Ferrand, Direction de la Recherche Clinique et de l’Innovation, F-63000 Clermont-Ferrand, France; clambert@chu-clermontferrand.fr (C.L.); bpereira@chu-clermontferrand.fr (B.P.); 5Université Clermont Auvergne, INSERM U1240, IMOST, F-63000 Clermont-Ferrand, France; emmanuel.moreau@uca.fr; 6Université Clermont-Auvergne, Unité INSERM 1107 Neuro-Dol, CHU Clermont-Ferrand, F-63000 Clermont-Ferrand, France; jpinguet@chu-clermontferrand.fr (J.P.); drichard@chu-clermontferrand.fr (D.R.)

**Keywords:** medical devices, PVC, plasticizers, exposure, neonatal intensive care unit, urinary metabolites

## Abstract

Care management of newborns in the neonatal intensive care unit (NICU) requires numerous PVC (PolyVinyl Chloride) medical devices (MD) containing plasticizers that can migrate and contaminate the patient. We measured the magnitude of neonates’ exposure to plasticizers (di-ethylhexylphthalate (DEHP) and alternatives) in relation to urinary concentrations of their metabolites. Plasticizers’ exposure was evaluated (1) by calculating the amounts of plasticizers prone to be released from each MD used for care management, and (2) by measuring the patients’ urinary levels of each plasticizers’ metabolites. 104 neonates were enrolled. They were exposed to di-isononylphthalate (DINP), especially via transfusion and infusion MD, and to DEHP via ECMO (Extra Corporeal Membrane Oxygenation) and respiratory assistance MD. Mean exposure doses exceeded the derived no-effect level of DINP and DEHP by a 10-fold and a 1000-fold factor. No PVC MD were plasticized with di-isononylcyclohexane-1,2-dicarboxylate (DINCH). High urinary concentrations of DEHP metabolites were directly correlated with DEHP exposure through ECMO MD. Urinary concentrations of DINP metabolites in transfused patients were also high. DINCH metabolites were found in urine, suggesting another route of exposure. Neonates in NICU are considerably exposed to plasticizers, with magnitudes varying with the type of MD used. The high exposure to DEHP and DINP leads to a risk of their metabolites’ toxicity.

## 1. Introduction

Current neonatal intensive care units (NICU) expose term and extremely low gestational age newborns to various plastic medical devices (MD). These MD include, for the most part, MD used for the administration of drugs, such as peripheral and central venous access catheters, infusion lines, feeding tubes and tubing, and airway management devices. Many of these MD are made of polyvinyl chloride (PVC) added with plasticizers used to impart flexibility and softness to PVC. The PVC contains different types and amounts of plasticizers depending on its application. It is now widely accepted that these plasticizers can migrate from the PVC matrix into drug solutions and thus come into contact with the patient [1,2,3].

Some of these plasticizers have been registered as at-risk compounds such as di-(2-ethylhexyl) phthalate (DEHP), which is now classed as a CMR1b (Cancerigen Mutagen or Reprotoxic) risk substance under the Classification Labeling and Packaging (CLP) Regulation [4], due to its effects on reproduction and fertility. The use of DEHP in PVC medical devices has been called into question by the European authorities and has been restricted since the last few years. It now must not exceed 0.1% by mass of the plasticized material, as defined by the European regulation n°2017/745 on medical devices [5].

DEHP is now well-known to induce adverse effects on the reproduction and development of humans, as it has been demonstrated in many epidemiological analyses, including children [6,7,8,9]. It is also classified as an endocrine disruptor by the European Chemicals Agency (ECHA) [10]. Indeed, recent studies show that a number of contaminants, including phthalates and particularly DEHP, disrupt several hormonal processes, such as steroid hormone and thyroid systems [11]. For example, DEHP can disrupt thyroid hormone signaling by reducing the circulating thyroid hormones (TH) levels, thus affecting growth, development, and differentiation, especially the developing brain [12,13]. It may also modify steroid hormone metabolism and balance by altering synthesis and/or breakdown of testosterone, follicle stimulation hormone, luteinizing hormone (LH), or other hormones involved in gamete physiology, fertility, implantation, fetal morphogenesis, pregnancy outcome, and post-birth diseases [14,15].

Moreover, fetal and neonatal periods are very sensitive periods to the effects of endocrine disruptors. Overall, multiple and intensive medical interventions and an impaired ability to excrete phthalates may explain the susceptibility of neonates, and particularly preterm infants, to high DEHP exposure, the latter being associated with the intensity and duration of invasive medical procedures [16,17,18,19,20,21,22].

Recently, MD manufacturers have started using some alternatives to DEHP plasticizers, especially for patient groups undergoing clinical procedures with high exposures (total parenteral nutrition, blood transfusion, oxygen therapy, ECMO (Extra Corporeal Membrane Oxygenation) and/or for high-risk patient groups such as neonates in NICU [23]. DEHP has been greatly replaced by chemicals such as di-(2-ethylhexyl) terephthalate (DEHT), di-isononyl cyclohexane-1,2-dicarboxylate (DINCH), and di-isononyl phthalate (DINP) in most of the MD used in NICU [24]. If some of them possess different physicochemical properties leading to a lower migration [2,23] and less risky toxicological profiles than DEHP [1], these alternative plasticizers are likely to present harmful effects. Data gaps exist for some molecules, whereas evidence of their endocrine disrupting potential was found, notably for DINP, which has already been banned from all toys and childcare articles [25]. Children’s exposure to DINCH should be investigated carefully as the total exposure may be close to the tolerable daily intake according to the National Industrial Chemicals Notification and Assessment Scheme (NICNAS) public report [26]. Moreover, Sheick et al. [27] recently observed that DINCH showed high binding affinity with sex hormone binging globulin (SHBG), suggesting that the alternate plasticizer has potential to strongly inhibit the steroid binding function of SHBG. Therefore, it cannot be excluded that DINCH can cause a potential disruption effect on the androgen and estrogen homeostasis of SHBG and thus interfere with steroid signaling functions. For DEHT, recent conclusions are that it is not clear whether this plasticizer causes reproductive toxicity or not [28]. However, as demonstrated by Kambia et al. [29], its oxidized metabolites seem to have an impact on sexual hormones, with an activity in the steroidogenesis assay, either in estradiol synthesis or in testosterone synthesis. Especially, one of the metabolites (5-OH-MEHT) showed a synergistic effect with the native ligand (17ßestradiol, E2) on human estrogen receptor α (hERα) and agonist effects on androgen receptors (ARs). Moreover, no data has been published on the exposure to these alternatives among critically ill neonates. Therefore, more and more pressing questions are arising about the potential endocrine-disrupting effect and long-term consequences on stature and neurocognitive development of these vulnerable newborns, which remain uncertain.

As the magnitude of exposure of neonates during their hospitalization in NICU to alternatives to DEHP plasticizers remains unknown, it is important to investigate their exposure levels. The aim of this prospective biomonitoring study was therefore to measure the magnitude of exposure of neonates and preterm newborns to plasticizers (DEHP and alternatives) and to quantitatively relate the use of plasticizers-containing products to urinary levels of plasticizers’ metabolites as biologic markers of this exposure.

## 2. Results

### 2.1. Study Population

A total of 104 neonates in both hospitals (63 boys, 41 girls) with a mean gestational age of 36.2 ± 4.3 weeks (27–44 weeks) and birth weight of 2.7 ± 0.9 kg were enrolled. Of the patients, 26/104 (25%) were followed until the end of the study (6th day). The details of patient characteristics are given in Appendix A.

As shown in Figure 1, the medical procedures performed in both centers, at day one, were statistically comparable. All patients received infusion therapies at day 1, and 96.2% of patients in center 1 and 100% of patients in center 2 needed respiratory assistance. Blood product transfusions were performed in 18 patients of center 1 and 15 of center 2. Three patients in center 2 required ECMO during respectively 5, 3, and 4 days. One patient in center 1 required plasmapheresis (at Day 1). No dialysis was performed.

### 2.2. Assessment of Plasticizers’ Exposure

All patients from both hospitals were exposed every day via MD to each plasticizer, except to DINCH (none of the MD contained DINCH), with a magnitude of exposure depending on the plasticizer and the hospital (Figure 2). The exposure to DEHT was very low, because DEHT was only found at trace levels in MD.

### 2.3. Assessment of Biomarkers in Urine

A total of 350 urinary samples were analyzed (208 from Center 1 and 142 from Center 2). All the metabolites of each plasticizer were found in the patients’ urine, but their levels varied widely, depending on the plasticizers and the center considered (Figure 3).

As expected, the concentrations of urinary DEHT metabolites were low compared to those of DEHP and DINP, as DEHT was found only at trace levels in MD used in this study.

On the contrary, unexpectedly, DINCH metabolites were found in all urine samples, suggesting another route of exposure than from MD.

We then focused on DEHP and DINP metabolites’ concentrations and their correlation with the two most exposing situations, i.e., ECMO (for DEHP exposure) and transfusion (for DINP exposure).

Figure 4 represents the correlation between DEHP exposure of each patient and their respective metabolites’ (sum of all metabolites) concentrations. It is clear that no correlation exists for all patients, except for the 3 patients undergoing ECMO, whose DEHP exposure doses and DEHP metabolites’ concentrations are far higher than all others. This observation is confirmed by statistical analyses when performing a multivariate regression, where the total exposure is significantly associated with the use of MD for ECMO (*p* < 0.001).

Additionally, the urinary concentrations of DEHP metabolites of the 3 patients undergoing ECMO were significantly much higher than those of the other patients in both centers (23,155.08 vs 640.27 ng/µmol creatinine; *p* = 0.003).

As shown on Figure 5, the exposure to DINP via transfusors is not correlated with urinary concentrations of DINP metabolites.

No significant differences were found between the urinary concentrations of DINP metabolites of transfused patients receiving at least one transfusion with a DINP-made MD and those of the other patients in both centers (20.90 vs 11.45 ng/µmol creatinine; *p* = 0.68)

For the patients of center 1 who were transfused, the median urinary level of Cx-MINP measured after transfusion was significantly higher than the level observed the day before (3.65 ng/µmol creatinine (2.26; 18.63) vs. 2.82 ng/µmol creatinine (0.49; 8.17) respectively, *p* = 0.02) (see details in Appendix A).

## 3. Discussion

Due to the rapidly growing worldwide regulations to limit the use of DEHP, other plasticizers are used as alternatives to soften the PVC of MD [30]. In NICU, due to the use of numerous and varied MD for the care management of neonates, their exposure to these alternatives is potentially high and continuous during their hospitalization.

Our study of 104 neonates is the first biomonitoring study assessing this exposure to alternative-to-DEHP plasticizers of neonates hospitalized in NICU. Previous NICU studies were conducted on few patients and were intended only to measure the exposure to DEHP via MD, based on the urinary concentrations of its metabolites [16,17,18,19]. All the authors demonstrated a correlation between the urinary levels of some DEHP metabolites (MEHP, 5-oxo-MEHP, and 5-OH-MEHP) and the exposure of NICU patients to DEHP via MD, higher than the general population. Moreover, Demirel et al. showed that the urinary excretion of DEHP metabolites was associated with the intensity of exposure to DEHP-containing products, the duration of exposure, and birthweight category of the patients [19].

Our study highlights that term and extremely low gestational age newborns in NICU are still exposed to DEHP despite the health concern relative to its proven endocrine disruption effect that has been taken into account by the European Authorities. In our study, all patients in both centers were exposed to DEHP. They were exposed to DEHP especially via MD used for respiratory assistance (centers 1 and 2) and ECMO (center 2). For respiratory assistance, while we found that most of the MD were made of DEHP plasticized PVC, exposure doses of patients were not high (compared to those of other clinical situations), which is probably linked to the lesser diffusion of DEHP from the MD into air compared to liquids, such as drug solutions or blood. However, the total estimated DEHP exposures in both centers remained high and were by far the highest among all plasticizers in center 2 due to the ECMO situation of 3 patients. Median values reached 0.30 (0.063–0.76) mg/day for center 1 and 22.48 (0.00; 490.44) mg/day for center 2. Considering a mean body weight of 2.7 kg for the patients in both centers, these values correspond to approximately 0.11 and 8.33 mg/kg/day, which are well above the derived no effect level (DNEL) of 0.05 mg/kg/day of DEHP as defined by ECHA [10]. The high variability obtained with DEHP exposure in center 2 is especially due to the specific procedure of ECMO, which was performed on only 3 patients whose exposure doses reached maximum values of 617.16 mg/day.

High concentrations of DEHP metabolites were found in urine of patients included in both study sites. The median value of the 5-Cx-MEHP concentrations of our studied patients from center 2 reached 487.12 (271.24; 1075.03) ng/µmol creatinine and 381.71 (102.49; 795.80) ng/µmol creatinine from center 1. Our results may be compared to those of Gaynor et al., for which 5-Cx-MEHP was measured before and after cardiac surgery in urine samples of 16 and 18 neonates, respectively [20]. The authors found a geometric mean of 1166.1 ng/mL of 5-Cx-MEHP (postoperative samples). This difference may be explained by the fact that their patients all underwent a cardiac surgery, requiring in 94% of cases a cardiopulmonary bypass, which is equivalent to the ECMO procedure given to only three patients in our study.

Expectedly, the three patients undergoing ECMO had levels of metabolites concentrations that were among the highest, especially for 5-Cx-MEHP, reaching 61,246 ng/µmol creatinine. However, as the ECMO was done according to the same procedure (machine setup, identical MD, and priming bag), it is clear that the metabolization process is variable from one neonate to another. This observation may be related to the growing appreciation that development during the critical period is particularly vulnerable to the effects of exogenous EDCs that can reprogram essential signaling/differentiation pathways and lead to lifelong consequences. As demonstrated by Kambia et al., the oxidized metabolites of DEHP and those of its alternatives may have a transcriptional effect on sexual hormone receptors, and may also impact the steroidogenesis synthesis, by increasing estradiol and decreasing testosterone. This effect is more pronounced for estradiol levels with 5-OH-MEHP, which is active from 0.2 µg/mL [29]. In our study, 5-OH-MEHP concentrations found in urine reached a maximum of 0.70 µg/mL in center 1 and 3.9 µg/mL in center 2, thus above the concentrations that induce this endocrine disruption action. By comparing the metabolites’ concentrations to the Reference Values 95 (RV95), statistically defined by the Human Biomonitoring Commission (HBM) to describe exposure to or body burden of environmental contaminants in the population at a given time, we found that 11.5% of our studied patients had a total concentration of both 5-OH-MEHP and 5-oxo-MEHP above the RV95 of 280 μg/L [31]. In addition, 5-Cx-MEHP concentrations are higher than the threshold of 200 µg/mL for 67.3% of the neonates, and with a five-fold increase for ECMO patients. 

Such results highlight that there is still a significant postnatal exposure to DEHP for neonates hospitalized in NICU. Furthermore, specific medical situations such as ECMO are at greatly increased risk of exposure (because of the modalities of the procedure and the use of numerous MD that are still plasticized with DEHP), leading to potential cytotoxic and endocrine disruption effects. Careful evaluation of whether this exposure can be reduced as far as reasonably possible should be performed [31]. 

We showed that neonates and premature newborns in NICU are exposed not only to DEHP but also to its alternatives used to soften the PVC of MD such as infusors, nutrition lines, extension lines, respiratory circuits, etc.

DINP is one of these alternatives, found mainly in transfusors for center 1 and infusion sets (infusors and extension lines) for center 2 in our study. Transfusion seems to be another high-risk medical situation, though a different way of exposure than ECMO, i.e., according to an intermittent but iterative process. Patients of our study were exposed for a short time per day (about 2 h) and for one or several consecutive days, which represent a much smaller period of exposure than ECMO. However, the exposure amount of DINP, especially in center 1 where DINP was the only plasticizer used in transfusion sets, may have reached 109.06 mg/day. Considering a mean body weight of 2.7 kg in both centers, this value corresponds to an exposure dose of 40.4 mg/kg/day, which is above the oral DNEL (4.4 mg/kg/day) defined by ECHA [10]. It should be noted that a comparison with a reference value given for intravenous exposure cannot be done because it does not exist, even if an average DNEL (79.9 mg/kg/day) of DINP has been proposed by Bui et al. [32]. Although a transfusion is a temporary and short-duration situation, patients might be highly exposed to plasticizers integrated in PVC MD, with urinary levels of metabolites increasing immediately (the day following the transfusion session). For center 2, the exposure to DINP is mainly mediated by MD of infusion and parenteral nutrition (PN) (and a small part via transfusion MD). Patients from center 2 are theoretically exposed to DINP at a level of 41.75 mg/day, corresponding to 15.46 mg/kg/day, four-fold higher than the oral DNEL. We did not show a significant correlation between the total exposure dose to DINP and the urinary levels of DINP biomarkers, nor by performing a multivariate analysis on the medical situation (i.e., transfusion and IPN (infusion and parenteral nutrition)). Contrary to the results found for the ECMO patients, concentrations of DINP metabolites in transfused patients receiving at least one transfusion with a PVC/DINP-made MD were not significantly higher than those of all other patients. This may be related to the fact that transfusion is not a clinical situation as exposing as the ECMO and thus the median values of DINP metabolites calculated on average per day are not high enough to make the difference with other patients (not transfused and transfused with a MD not made in PVC/DINP) significant.

However, the levels of DINP metabolites (3.17 ± 4.72 ng/µmol creatinine (center 1) and 8.89 ± 17.48 ng/µmol creatinine (center 2) for oxo-MINP, 6.62 ± 6.58 ng/µmol creatinine (center 1) and 15.36 ± 25.73 ng/µmol creatinine (center 2) for OH-MINP, and 21.64 ± 32.52 ng/µmol creatinine (center 1) and 32.63 ± 57.47 ng/µmol creatinine (center 2) for Cx-MINP) are slightly higher than those found in adults in the work of Koch et al. [33] (14.9 ng/mL for OH-MINP, 8.9 ng/mL for oxo-MINP, and 16.4 ng/mL for Cx-MINP) and in the same range as those found by Lin et al. [34] in the urine of two-year-old children (mean of 27.8 ng/mL), suggesting the higher risk of exposure of the NICU population. Moreover, 5.8% of the studied patients had a sum of the concentrations of the three oxidized DINP metabolites higher than the RV95 value of 140 µg/L, which is calculated for children between 3 and 14 [31]. Precisely, 7.7% of the patients had Cx-MINP rates higher than the threshold of 60 µg/L, and oxo-MINP and OH-MINP exceed the respective values of 30 and 50 µg/L in 3.8% and 4.3% of patients’ urine.

To date, no endocrine disruption effect has been assessed with DINP metabolites, but recent data shows that the parent plasticizer (DINP) exhibits a binding affinity with SHBG in the in silico study of Sheikh et al. [27], with a docking complex that forms 86% of interactions. Moreover, DINP exposure led to significant transcriptional changes even at low-exposure concentrations, suggesting a great endocrine disruption potency when tested in zebrafish in the work of Lee et al. [35].

DEHT seems to be an interesting alternative to DEHP. In our study, the exposure of neonates to DEHT is very low, because it was not integrated as a major plasticizer in the composition of MD. The low levels of its metabolites found in urine reflect this poor exposure, being marginal compared to the doses that have been shown as being able to induce an agonist or an antagonist effect on androgen or estradiol receptors (beginning at 20 ng/mL for 5-OH-MEHT) and without any cytotoxic effect on HeLa-9903 and MDA-kb2 cells for the monoester nor for the corresponding oxidized metabolites, as demonstrated by Kambia et al. [29]. Several elements are playing in favor of DEHT as an alternative to DEHP: weak diffusion towards the liquids in contact with the medical devices, limiting the exposure of the patients [2], and less toxicity compared to the DEHP (cytotoxicity, reprotoxicity, endocrine disruption) [28,29].

Theoretically, our studied patients should not have been exposed to DINCH, at least not via MD. As we found trace levels of its metabolites in the urine, we suggest that the contamination is provided by another source, such as incubators (by inhalation exposure, another route of exposure to plasticizers [36,37,38]), babies’ diapers [39,40,41], or the mother-to-child transmission during pregnancy (this latter being quite unlikely because of the fast elimination of DINCH metabolites in the organism [42,43], while we found levels in the urine of the babies each study day), or even during breastfeeding.

Additionally, most NICU inpatients have multiple sources of exposure to different plasticizers concurrently (i.e., those receiving respiratory support were likely also be receiving some sort of nutrition). This observation may be of concern regarding the recent data showing that the exposure effects of a mixture of several contaminants such as endocrine disruptors may enhance behavioral effects in animal [44,45] or human cells [46].

Finally, our study highlights that neonates’ exposure is a function of each plasticizer and on the center, suggesting different methods/medical procedures used for the care management of those patients (time of performing PN simultaneously with EN, infusion modalities in terms of flow rates, duration, types of drugs administrated, frequency of MD replacement, etc.). For DINP and DEHP, the plasticizers causing most of the exposure, patients of center 2 were far more exposed to DINP and DEHP than those of center 1, respectively 7.5 times and 75 times more. For DEHP, this higher exposure was caused by MD used for ECMO therapy in center 2, which was not realized in center 1. The exposure to DINP in center 1 affected exclusively patients who received blood transfusion, whereas the most part of this exposure in center 2 was provided by MD infusion and nutrition MD. Another explanation for the differences in plasticizers exposure may be provided by the public contracts for the supply of MD. Each hospital is free to buy its own MD references, according to its institutional policy. Thus, as reported in the work of Bourdeaux et al., the choice of the plasticizer(s) included in PVC MD is exclusively imposed by the supplier. Indeed, currently, and contrary to the regulations for Food Contact Materials, the harmonized standards (EN ISO 10993) for the biological assessment of MD do not impose the nature nor acceptable migration limits for additives (plasticizers) incorporated into these specific non-invasive class I or invasive MD that do not come into direct contact with the vascular system and are intended for temporary use. The European Regulation N°2017-45 introduces for the first time a quantitative restriction for the use of additives in MD that are classified CMR (Cancerigen, Mutagen, Reprotoxic) 1A or 1B and/or have a proven endocrine disruption effect. Actually, only DEHP (among all plasticizers integrated in PVC MD) is affected by this restriction (the amount must be under 0.1% of the mass fraction of the MD). 

The Fick’s law model we used to quantify the theoretical exposure amounts took into consideration the apparent diffusion coefficient of each plasticizer, which was calculated within experimental conditions using the validated hydro-alcoholic simulant (ethanol/H20 50/50 (v/v) [47]). Consequently, it might have slightly overestimated the exposure doses of the NICU patients because of the conditions in which those experiments were conducted. Indeed, MD samples (cut pieces) were in contact with high volumes of simulant during a long time, leading to low transfer resistance and thus large amounts of plasticizers released, especially in the first minutes. Moreover, the apparent diffusion coefficient was determined at the steady state. These conditions are likely reflecting the worst-case scenario of the release of the plasticizers into simulant.

## 4. Materials and Methods

### 4.1. Study Population

This biomonitoring study was conducted as a part of the ARMED project (Assessment and risk management of medical devices in plasticized polyvinyl chloride), which received the financial support of the French Medicine Agency (ANSM, Agence Nationale de Sécurité du Médicament et des Produits de Santé). 

The clinical trial part of the ARMED project was an observational study with biological collection and associated data, conducted between April 2014 and May 2015. It was approved for all centers by a central ethics committee (Comité de Protection des Personnes Sud-Est VI, Clermont-Ferrand, France) and the collection was registered to the French Ministry of Higher Education and Research. It was identified in the European Clinical Trials Database (EudraCT 2015-002559-84) and at ClinicalTrials.gov (NCT02618720). 

After both parents or a legal representative gave their non-opposition, patients admitted to the level 3 NICUs of 2 French university hospitals (center 1 = coordinator site and center 2 = associated site) were enrolled if (1) it seemed likely that their hospital length would be ≥2 days, (2) they had a urinary catheter, and (3) they were submitted to at least one of the following medical procedures: enteral nutrition (EN), parenteral nutrition (PN), hemodialysis (HD), extracorporeal membrane oxygenation (ECMO), extracorporeal situation during a cardiac surgery, blood exchanges transfusion. They were included for a maximum period of 6 days.

Exclusion criteria were (1) patient’s death, (2) removal of the urinary catheter, and (3) patients becoming anuric.

### 4.2. Study design

The design of the study is presented in Figure 6. 

Each day, the following data and samples were collected:

In a case report form (CRF), all PVC MD used for the care management of the patients were inventoried and registered for: number, date, and duration of the use (see an extract in the Appendix A).

Once a day at 8:00 a.m., a urine sample was withdrawn from a non-PVC bottle into which the collection bag used to collect the standard urinary draining was emptied 6 times a day for each patient. This collected urine sample therefore reflects the amounts of metabolites released during the previous 24 h.

### 4.3. Assessment of Plasticizers Exposure

In order to estimate each patient’s level of exposure to the plasticizers, we first evaluated the amounts of plasticizers prone to be released from each MD reported in the CRF into the fluid in contact with it. To this end, we used a Fick’s law-based model (adapted from the Fick’s diffusion law [48], Appendix A):M = D × C × 1000 × 1.27 × S × t(1)

M: quantity of plasticizer released from the MD (mg),

C: plasticizer concentration within the MD, in g per g of PVC (obtained by Gas Chromatography-Mass Spectrometry (GC-MS) according to Bourdeaux et al. [49]),

D: apparent plasticizer diffusion ability (cm/min), determined through experimentation (Appendix A),

1.27: PVC density,

S: PVC area which is in contact with the drug solution or biological fluid (cm²),

t: duration of exposure to the medical device (min).

For MD used for respiratory assistance, from which the plasticizers are released into a gaseous media, the Fick model might not be appropriate and could therefore lead to exposure overestimation. In recent work (development of a respiratory assistance ex vivo model [50]), our team showed that the migration of plasticizers into gas is about 3000 times lower than into liquid media, such as drug solutions. This observation is supported by several other studies, all showing that DEHP migration into air is between 500 and 20,000 times lower than into a liquid media [51,52,53,54,55]. Consequently, we applied a corrective factor of 3000 to estimate the amounts of plasticizers that newborns requiring respiratory support are exposed to during their hospitalization.

Finally, we calculated the daily quantity of each plasticizer to which each patient had been exposed to according to the MD used for his care management in NICU.

### 4.4. Assessment of Biomarkers in Urine

The daily urine sample collected was stored in 6.5 mL Vacuette^®^ Z urine tubes with no additives (Greiner Bio-One, France), immediately shipped to the Center of Biological Resource (CBR) on each site and kept frozen at −20 °C until analysis. We validated beforehand that the entire sampling procedure did not induce any sample contamination by the studied plasticizers.

Finally, all samples from center 2 were shipped to the coordinator site for the analysis. The urine specimens of both sites were analyzed for DEHT, DINP, DINCH and DEHP metabolites (Figure 7) by the automated online SPE-LC-MS/MS (Solid Phase Extraction-Liquid Chromatography- Mass Spectrometry/MassSpectrometry) method developed by Pinguet et al. [56], after a sample hydrolysis using β-glucuronidase. The urinary concentrations of biomarkers were expressed in ng/mL and then creatinine-adjusted (ng/µmol creatinine).

### 4.5. Statistical Analysis

Statistical analysis was performed using Stata software (version 13; StataCorp, College Station, TX, USA). All tests were two-sided, with type I error set at 0.05. Categorical parameters were expressed as frequencies and associated percentages, and continuous data as mean ± standard deviation or as median (interquartile range), according to the statistical distribution. For each patient, the exposure to plasticizers was assessed in two ways: exposure on the first day and daily average exposure (sum of all exposures divided by the length of stay in days). At baseline, the patients of the two centers were compared using the chi-squared test or the Fisher’s exact test (for categorical parameters) and the Student’s t test or Mann–Whitney test (for quantitative parameters), as appropriate. The Gaussian distribution was verified by the Shapiro–Wilk test and homoscedasticity by the Fisher–Snedecor test. To assess the relationship between exposures to plasticizers, and the amounts of plasticizers and metabolites found in the urine, we used the Spearman’s rank correlation coefficient for each plasticizer (ranging from −1 (a perfect negative correlation) and 1 (a perfect positive correlation)). Linear mixed regressions were also implemented (considering the center as a random effect) in order to see the influence of the type of MD on the amounts of metabolites found in the urine. In these models, the dependent variable was the amounts of metabolites found in the urine (for example DEHP metabolites) and the independent variables were the (DEHP) exposure due to each type of MD (ECMO, EN, infusion and PN, respiratory assistance, transfusion). Finally, the concentrations of Cx-MINP (the most specific biomarker) before and after transfusion sessions were compared using linear mixed models, considering the patient as a random effect (because the same patient could have several transfusion sessions) and the time (before/after transfusion session) as an independent variable.

## 5. Conclusions

We demonstrated that neonates were significantly exposed to DEHP and alternative plasticizers that are contained in the MD used for their care management in NICU, i.e., drug administration. Moreover, some medical procedures and some plasticizers are at higher risk of exposure. ECMO patients are highly exposed to DEHP, for which a direct association with urinary metabolites may be established. High DINP exposure levels are also caused by transfusion sessions performed on neonates, with values above the DNEL fixed by ECHA. DINCH rates found in urine of the patients without any theoretical exposure are still a matter for debate, and raise the question of an unavoidable contamination and toxic risk of this sensitive population. Nevertheless, due to the concern about DINP and DINCH toxicity, both plasticizers do not represent safe candidates to replace DEHP in the MD. Pharmaceutical firms should choose plasticizers with low migration from PVC matrix and low toxicity, such as DEHT. However, more research is needed to determine whether neonates who undergo intensive therapeutic interventions and receive numerous infused drugs using only DEHT-containing devices are at higher risk for altered health outcome (i.e., related to the endocrine disruption effect to which they are more sensitive) than those who undergo similar treatments but are exposed simultaneously to a mixture of plasticizers.

## Figures and Tables

**Figure 1 metabolites-11-00252-f001:**
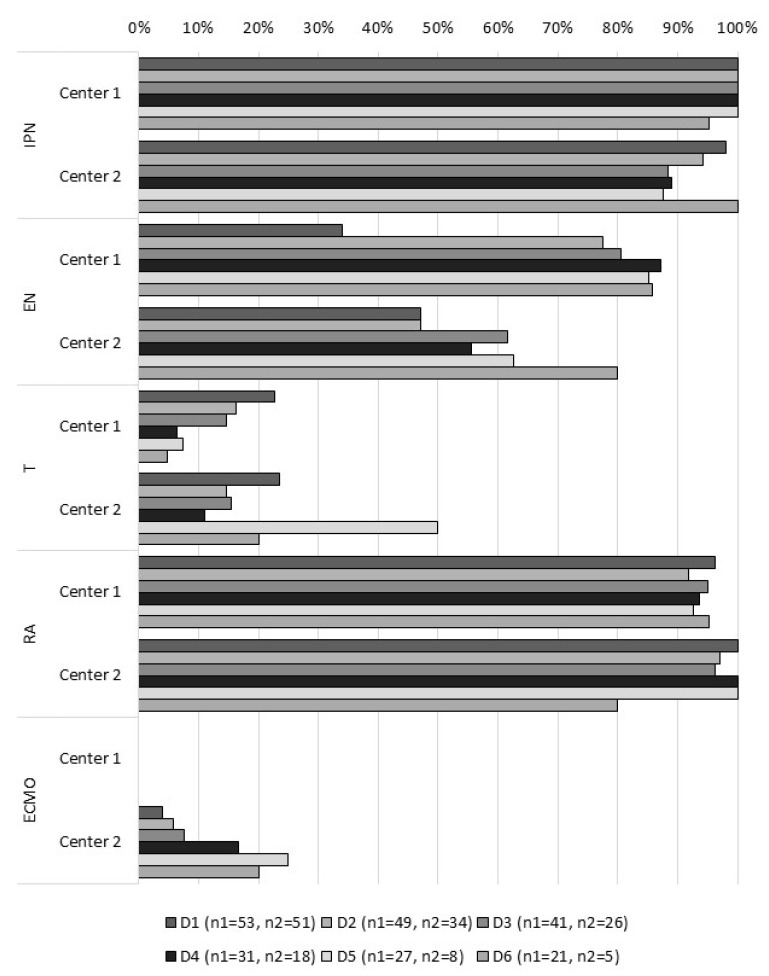
Percentage of patients within the 6 days of the study, according to the medical procedure (n1 = number of patients in center 1; n2 = number of patients in center 2). IPN: infusion and parenteral nutrition; EN: enteral nutrition; T: transfusion; RA: respiratory assistance; ECMO: extracorporeal membrane oxygenation.

**Figure 2 metabolites-11-00252-f002:**
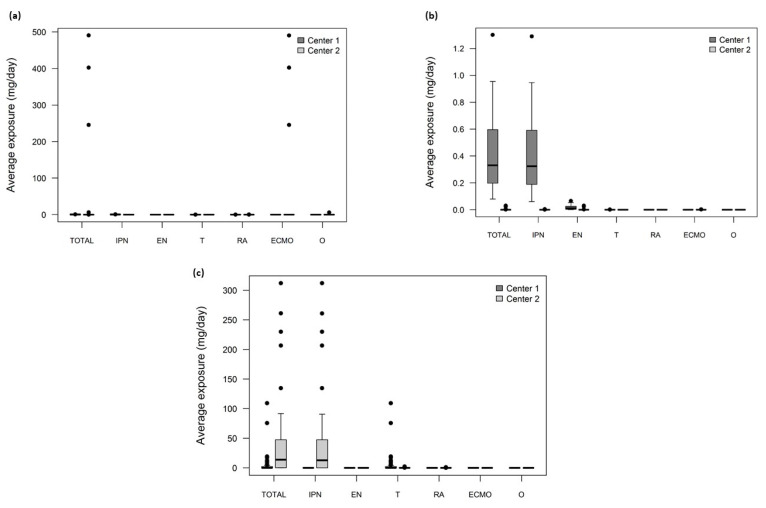
Boxplot of the average exposure doses of patients (mg/day) calculated by Fick’s model for: (**a**) DEHP, (**b**) DEHT, and (**c**) DINP (the central mark is the median, the edges of the box are the 25th and 75th percentiles, the upper whisker is calculated as the maximum of (75th percentile + 1.5 × (75th percentile – 25th percentile)), the lower whisker is calculated as the minimum of (25th percentile–1.5 × (75th percentile – 25th percentile)), and the dots represent outliers. IPN: infusion and parenteral nutrition; EN: enteral nutrition; T: transfusion; RA: respiratory assistance; ECMO: extracorporeal membrane oxygenation; O: others. DEHP: di-(2-ethylhexyl) phthalate; DEHT: di-(2-ethylhexyl) terephthalate (DEHT); DINP: di-isononyl phthalate.

**Figure 3 metabolites-11-00252-f003:**
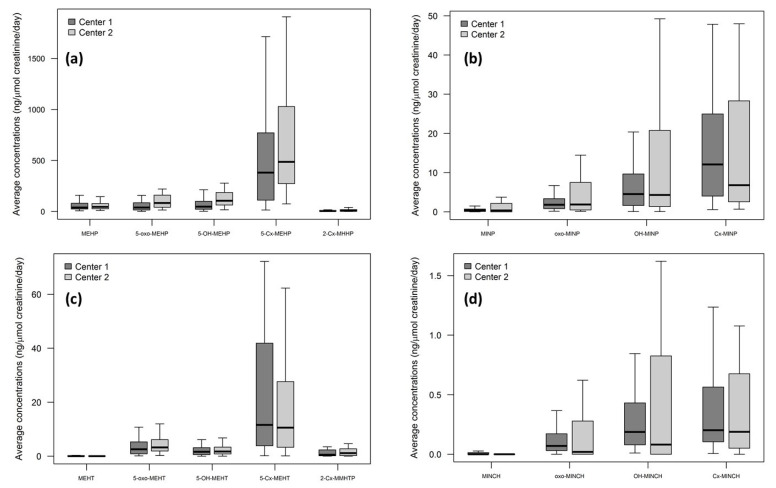
Boxplot of the average concentrations (ng/µmol creatinine/day) of urinary plasticizers metabolites for: (**a**) DEHP, (**b**) DINP, (**c**) DEHT, and (**d**) DINCH. For clarity, outside values were not presented. DEHP and DINP metabolites were found at much higher levels than those of DINCH and DEHT. As shown in Table 1, there was no correlation between DEHP or DINP metabolites’ concentrations and the respective exposures to DEHP or DINP when considering the total sites exposure. DEHP: di-(2-ethylhexyl) phthalate; DEHT: di-(2-ethylhexyl) terephthalate (DEHT); DINP: di-isononyl phthalate; DINCH: di-isononyl cyclohexane-1,2-dicarboxylate.

**Figure 4 metabolites-11-00252-f004:**
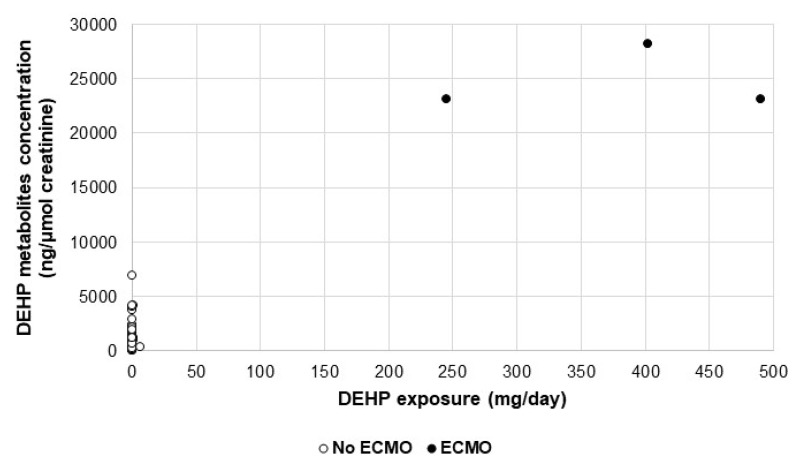
Scatterplot of DEHP metabolites’ concentration by DEHP exposure, according to the presence (black dots) or absence (white dots) of ECMO procedure.

**Figure 5 metabolites-11-00252-f005:**
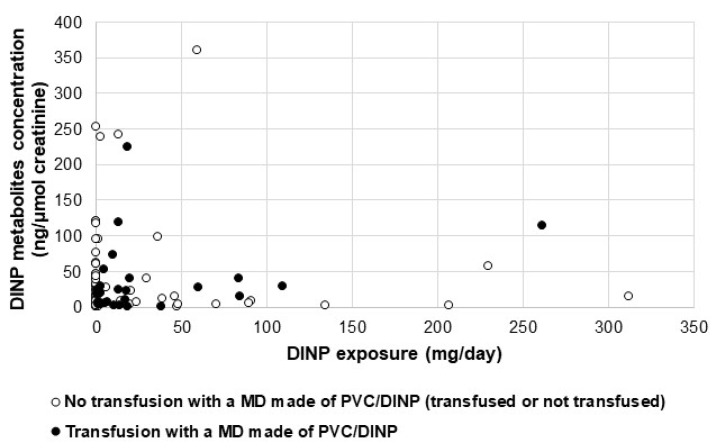
Scatterplot of DINP metabolites’ concentration by DINP exposure, according to the presence (black dots) or absence (white dots) of at least one transfusion performed with a MD made of PVC (PolyVinyl Chloride)/DINP.

**Figure 6 metabolites-11-00252-f006:**
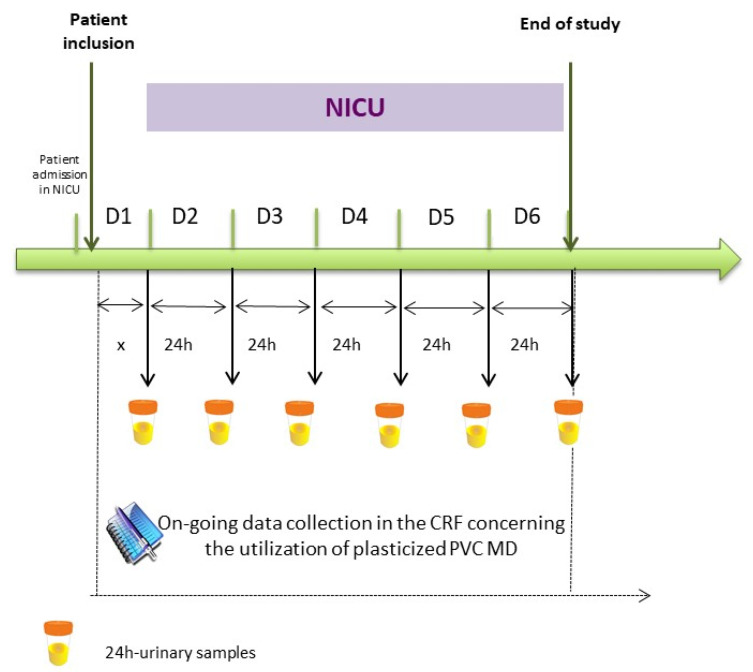
ARMED study design. CRF = Case Report Form. MD = Medical Device. ARMED project: Assessment and risk management of medical devices in plasticized polyvinyl chloride).

**Figure 7 metabolites-11-00252-f007:**
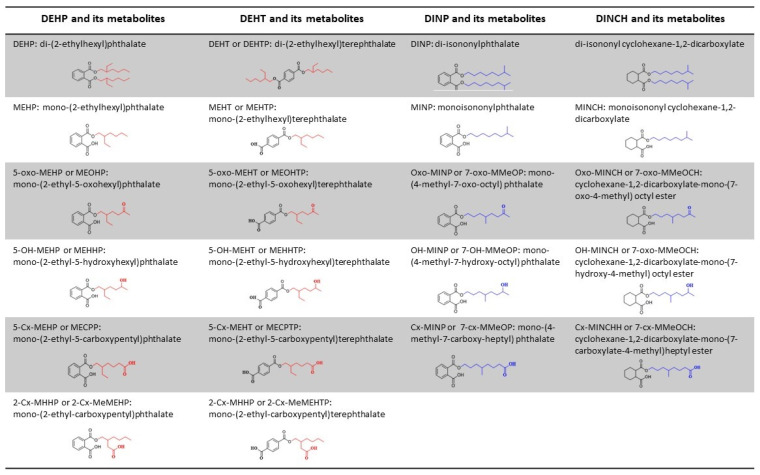
Structure and denomination of DEHP, DEHT, DINP, and DINCH, and their metabolites.

**Table 1 metabolites-11-00252-t001:** Correlations between measurements of urinary plasticizers’ metabolites and dose exposure to plasticizers in both hospitals of the study.

	Center 1 (n = 51)		Center 2 (n = 51)		Total (n = 102)	
	ρ	*p*	ρ	*p*	ρ	*p*
DEHP	0.18	0.22	0.04	0.77	0.15	0.13
DEHT	0.20	0.16	0.15	0.31	0.08	0.42
DINP	−0.06	0.65	0.09	0.54	0.03	0.74

ρ: Spearman’s rank correlation coefficient; *p*: *p*-value.

## Data Availability

Data is contained within the article.

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
