# Peer review of "Association between Urinary Metabolites and the Exposure of Intensive Care Newborns to Plasticizers of Medical Devices Used for Their Care Management"

_metabolites, 2021, doi:10.3390/metabo11040252_

Round 1

Reviewer 1 Report

The manuscript by Lise Bernard et al entitled “Association between urinary metabolites and the exposure of intensive care newborns to plasticizers of medical devices used for their care management” reports evaluation of the plasticizers’ exposure in neonatal ICU patients with PVC medical devices application. They assessed urinary concentrations of the plasticizers (di- 21 ethylhexylphthalate (DEHP) and alternatives) of their metabolites. High amount of exposure doses are identified as urinary DEHP metabolites through ECMO application and DINP metabolites in transfused patients. Overall, it is a interesting study.

A table to summarize the patient characteristics may be included as a supplementary data.

Author Response

Comments and Suggestions for Authors

The manuscript by Lise Bernard et al entitled “Association between urinary metabolites and the exposure of intensive care newborns to plasticizers of medical devices used for their care management” reports evaluation of the plasticizers’ exposure in neonatal ICU patients with PVC medical devices application. They assessed urinary concentrations of the plasticizers (di- 21 ethylhexylphthalate (DEHP) and alternatives) of their metabolites. High amount of exposure doses are identified as urinary DEHP metabolites through ECMO application and DINP metabolites in transfused patients. Overall, it is a interesting study.

A table to summarize the patient characteristics may be included as a supplementary data

Response : A supplementary file has been added, providing the detailed characteristics of the patients enrolled in the study. All the suppementary files have been renumbered in the right order

Reviewer 2 Report

Interesting study on the exposure of neonates admitted to the neonatal ICU to plasticizers whose origin is medical devices used (intravenous lines, respiratory supports,…).

Conclusion indicates that plasticizers are dependent on the devices used and that exposure to DEHP and DINP is possibly of concern.

The article is relevant and of sufficient quality to be published, with minimal changes.

Summary: correct.

Introduction: adequate and updated review of the topic.

Results: I miss an assessment by gestational age and / or weight of the patient, given their great variability.

Discussion: adequate.

Materials and methods: adequate. There is a possible error in the numbering of the figures (figures 1, 2 and 3)

Conclusions: correct.

References: 24 and 25 are repeated.

Author Response

Interesting study on the exposure of neonates admitted to the neonatal ICU to plasticizers whose origin is medical devices used (intravenous lines, respiratory supports,…).

Conclusion indicates that plasticizers are dependent on the devices used and that exposure to DEHP and DINP is possibly of concern.

The article is relevant and of sufficient quality to be published, with minimal changes.

Summary: correct.

Introduction: adequate and updated review of the topic.

Results: I miss an assessment by gestational age and / or weight of the patient, given their great variability.

Response : Detailed characteristics of the patients enrolled have been given in a new supplementary file. However, this requested assessment would need numerous statiscal analyses and will be the subject of another article entirely apart. This article (which is ongoing) will assess the impact of the age, weight and environmental factors on the patients’ exposure to plasticizers and the urinary distribution of all types of their respective metabolites.

Discussion: adequate.

Materials and methods: adequate. There is a possible error in the numbering of the figures (figures 1, 2 and 3)

Response : All the figures have been renumbered in the right order, both within the main text and the figures’ captions

Conclusions: correct.

References: 24 and 25 are repeated

Response : The reference 25 was a mistake, sorry. It has been corrected.